# Bee Venom Proteins Enhance Proton Absorption by Membranes Composed of Phospholipids of the Myelin Sheath and Endoplasmic Reticulum: Pharmacological Relevance

**DOI:** 10.3390/ph18091334

**Published:** 2025-09-05

**Authors:** Zhuoyan Zeng, Mingsi Wei, Shuhao Zhang, Hanchen Cui, Ruben K. Dagda, Edward S. Gasanoff

**Affiliations:** 1Advanced STEM Research Center, Beijing Chaoyang Kaiwen Academy, Beijing 100018, China; 1704030012@cy.kaiwenacademy.cn (Z.Z.); 2108260100@cy.kaiwenacademy.cn (M.W.); 1806040018@cy.kaiwenacademy.cn (S.Z.); 2108270082@cy.kaiwenacademy.cn (H.C.); 2Department of Pharmacology, University of Nevada Medical School, Reno, NV 89557, USA; rdagda@med.unr.edu; 3Belozersky Institute of Physico-Chemical Biology, M.V. Lomonosov Moscow State University, Moscow 119991, Russia

**Keywords:** bee venom proteins, surface-mediated proton absorption, bioenergetics, phospholipid membranes, myelin sheath

## Abstract

**Background/Objectives**: Recent evidence challenges the classical chemiosmotic theory, suggesting that proton movement along membrane surfaces—not bulk-phase gradients—drives bioenergetic processes. Proton accumulation on membranes like the myelin sheath and endoplasmic reticulum (ER) may represent a universal mechanism for cellular energy storage. This study investigates whether phospholipids from these membranes, combined with anionic bee venom proteins, enhance proton absorption, potentially elucidating a novel bioenergetic pathway. **Methods**: Five phospholipids (phosphatidylethanolamine, phosphatidylserine, phosphatidylinositol, sphingomyelin, phosphatidylcholine) from rat liver were isolated to model myelin/ER membranes. Anionic proteins (p*I* 5.65–5.80) were purified from bee venom via cation exchange chromatography. Liposomes (with/without proteins) were prepared, and proton absorption was quantified by pH changes in suspensions versus pure water. Statistical significance was assessed via ANOVA and *t*-tests. **Results**: All phospholipid liposomes examined in this study absorbed protons under the tested conditions, with phosphatidylethanolamine showing the highest capacity (pH increase: 7.00 → 7.18). Liposomes enriched with anionic proteins exhibited significantly greater proton absorption (e.g., phosphatidylserine + proteins: pH 8.15 vs. 7.15 alone; *p* < 2.43 × 10^−6^). Sphingomyelin-protein liposomes absorbed the most protons, suggesting that protein–phospholipid interactions modulate surface proton affinity. **Conclusions**: Anionic bee venom proteins amplify proton absorption by phospholipid membranes, supporting the hypothesis that lipid–protein complexes act as “proton capacitors”. This mechanism may underpin extramitochondrial energy storage in myelin and ER. Pharmacologically, targeting these interactions could mitigate bioenergetic deficits in aging or disease. Further research should define the structural basis of proton capture by membrane-anchored proteins.

## 1. Introduction

The chemiosmotic theory, which proposes a mechanism for mitochondrial ATP production, was first introduced by Peter Mitchell in 1961 [1]. The key element of chemioosmotic theory is the oxidation reduction reaction that creates a concentration gradient of protons across the inner mitochondrial membrane (IMM)—a fundamental necessity that drives ATP synthesis by mitochondrial ATP synthase. Mitchell’s chemiosmotic theory revolutionized our understanding of oxidative phosphorylation in bioenergetics, earning him the Nobel Prize in Chemistry in 1978. In addition to its fundamental contributions to biochemistry, this theory has catalyzed significant breakthroughs in biomedical research—especially in pharmaceutical development, where it serves as the cornerstone for therapies designed to target mitochondrial dysfunction and associated energy production deficits [2]. According to this theory, proteins of the electron transport chain in the inner mitochondrial membrane (IMM) transfer electrons, releasing energy that is used to pump H^+^ ions from the mitochondrial matrix into the intermembrane space. This process creates an electrochemical gradient across the IMM. To dissipate this gradient, H^+^ ions flow back into the matrix through the proton channel of ATP synthase, driving the rotation of the Fo subunit, resulting in ATP synthesis by the F_1_ subunit [3].

Although the chemiosmotic theory has gained broad acceptance, it presents several unresolved issues. A major concern raised by many researchers is the proposed accumulation of H^+^ ions in the bulk water of intermembrane space and their depletion in the matrix [4,5,6,7]. Such changes would alter the pH of bulk water on both sides of the IMM, potentially creating unphysiological conditions in the mitochondrion [8,9]. To address this, scientists from various countries have proposed that phospholipids may facilitate the transport of H^+^ ions along the surface of the IMM directly to ATP synthase [10,11,12,13,14,15,16,17]. This surface-based transport would generate an electric potential localized to the membrane interface rather than within the bulk aqueous phases [14,16,18,19]. The resulting potential is thought to produce hydrodynamic forces that rotate the catalytic domain of ATP synthase, leading to ATP release [11,12,13,14,15,16,17,18,19]. Furthermore, it has been demonstrated that the rate of ATP synthesis is enhanced by proton movement along the membrane’s surface [20,21], rather than by a proton concentration gradient in the bulk water [22,23,24,25,26].

Innovative work of Morelli group provided evidence of extramitochondrial (in absence of mitochondria) energy production. Morelli et al. showed that the myelin sheath devoid of mitochondria and surrounding nerve cells is capable of synthesizing ATP [27], which is then delivered to axons to support electrical signal propagation in the central nervous system [28]. The same group further showed that ATP can be synthesized in other systems—such as plasma membranes, optic nerve myelin, platelets, microvesicles, exosomes, and the endoplasmic reticulum (ER)—even in the absence of mitochondria or a transmembrane H^+^ gradient [29,30,31,32,33,34]. It was proposed that any membrane incorporating electron transport chain proteins and ATP synthase—possibly via ER-mediated transfer from mitochondria [35,36] or other means [37,38,39]—can synthesize ATP, provided that it can absorb and retain protons, functioning as a “proton capacitor” to store the energy needed for ATP production [40,41,42,43]. The term “proton capacitor” is used explicitly as a metaphorical framework used by Morelli and colleagues, while acknowledging that the underlying phenomenon—surface proton accumulation as a form of energy supply—is supported by the broader literature [44]. Related concepts of proton trapping and lateral proton conduction have been documented by other groups [10,11,13,14,15,16,45], supporting the broader relevance of this framework.

Although the binding and movement of protons on surface of mitochondrial membranes with anionic cardiolipin—the major phospholipid in IMM that absorbs the protons—has been well documented [2,3,8,9,11,45,46,47,48], the major phospholipids that absorb protons on surface of plasma membrane and other extramitochondrial energy-producing membranes have not been identified. In this study, we isolated phospholipids from rat liver—phosphatidylethanolamine, phosphatidylserine, phosphatidylinositol, sphingomyelin, and phosphatidylcholine—which are commonly found in the membranes of the myelin sheath and ER. While isolated phospholipids do not capture the full complexity of native myelin or ER membranes, they provide a simplified and well-established model system for probing fundamental lipid–protein interactions. We hypothesize that anionic phospholipids, such as phosphatidylserine and phosphatidylinositol, have a higher capacity for absorbing the protons and acting as proton capacitors. We have also isolated anionic proteins from bee venom and used them to prepare liposomal membranes composed of the aforementioned phospholipids. By using traditional physicochemical methodology, we investigated whether these anionic proteins enhance the ability of such membranes to absorb protons from the bulk aqueous phase, a property that may be relevant to the function of membranes in the myelin sheath and ER as proton capacitor. Overall, our results demonstrate that all liposomal membranes absorb protons from bulk water, and that liposomes enriched with anionic proteins exhibit greater proton absorption than their protein-free counterparts. These findings underscore the capacity of lipid–protein membranes to accumulate protons at their surface, supporting a revised view of membrane-associated energy storage.

## 2. Results

We fractionated the whole bee venom by cation-exchange CM Sephadex C-50 column chromatography, as described in the Materials and Methods section, in an attempt to isolate anionic and cationic proteins that supposedly evolved from common cellular membrane-active proteins [49,50,51]. Optical density values of the eluants collected in 40 tubes measured at 280 nm are given in Table 1. A chromatogram reflecting the order of proteins coming out with eluants in the cation-exchange column is shown in Figure 1.

The KCl gradient buffer starts from tube 23. As the concentration of K^+^ ions increases, the cationic proteins exchange with K^+^ ions at the binding sites of the CM Sephadex C-50 resin and start coming out of the column at tubes 28 and 29 (Figure 1). Overall, the tubes were pooled into the seven fractions, as indicated in Figure 1.

The SDS-PAGE electrophoretic run reveals the molecular weights of proteins in fractions 1–7 ranging from about 12 kDa to 44 kDa (Figure 2). Melittin, the major protein in bee venom with a molecular mass of 2.8 kDa, is not shown in Figure 2 because it diffused out of the 3.5 kDa cutoff bag during overnight dialysis. IEF run represents the protein isoforms in each protein fraction, as proteins in each fraction have slightly different p*I* values (Figure 3). Notably, protein isoforms in each fraction have same molecular masses (Figure 2). For example, proteins in fraction 1 all have the same molecular mass of about 12.5 kDa, but their p*I* values range from about 5.65 to 5.80. The IEF run confirms that proteins in fractions 1– 5 are anionic, with p*I* values from 5.65 to about 6.3, while proteins in fractions 6 and 7 are cationic, with p*I* values from 8.3 to about 10.5. For preparation of phospholipid liposomes enriched with proteins, we selected fraction 1, containing the most anionic proteins, with p*I* values ranging from about 5.65 to 5.80.

The pH values in dd-H_2_O and in the liposomes of various phospholipids are shown in Table 2. Although the concentration of phospholipids was very low (10^–5^ M), the liposomes did increase pH values with the highest increase observed in liposomes made of phosphatidylethanolamine. The low *p*-value of the ANOVA test indicates that the difference in pH values within samples of dd-H_2_O and liposomes is statistically significant.

The difference between pH values in liposomes made of only phospholipids and liposomes made of phospholipids and fraction 1 proteins is shown in Table 3. The low *p*-values from the *t*-test indicate that the difference in pH values between liposomes made of only phospholipids and liposomes made of phospholipids and fraction 1 proteins is statistically significant. One can see that the liposomes made of phospholipids and fraction 1 proteins have the higher pH values than liposomes made of only phospholipids.

The concentrations of H^+^ ions absorbed by fraction 1 proteins in dd-H_2_O, by liposomes made of only phospholipids, and by liposomes made of phospholipids and fraction 1 proteins—which was calculated as the difference between initial concentration of H^+^ ions in pure dd-H_2_O and final concentration of H^+^ ions when fraction 1 proteins or liposomes are added to dd-H_2_O—is shown in Table 4. Interestingly, fraction 1 proteins absorb more H^+^ ions than liposomes made of any type of phospholipids; however, liposomes made of phospholipids and fraction 1 proteins absorb more H^+^ ions than fraction 1 proteins in dd-H_2_O. Except for liposomes made of sphingomyelin and fraction 1 proteins, the liposomes made of phospholipids and fraction 1 proteins absorb less H^+^ than the sum of H^+^ ions absorbed together by fraction 1 proteins in dd-H_2_O and by liposomes made of only phospholipids. This may be due to the insertion of fraction 1 proteins into membranes of phospholipid liposomes, which reduces the surface area of fraction 1 proteins exposed to the bulk water, resulting in reduced proton absorption. In cases of liposomes made of sphingomyelin and fraction 1 proteins, the packing of sphingomyelin and proteins in membrane disturbs the bilayer packing of sphingomyelin, resulting in the exposure of polar head groups of sphingomyelins to buck water, leading to an increase in the concentration of absorbed H^+^. Figure 4 compares the values of the negative logarithm of concentration of H^+^ ions absorbed by liposomes made of phospholipids with that of liposomes made of phospholipids and fraction 1 proteins. The lower the negative logarithm value, the higher the concentration H^+^ ions absorbed.

One can see that the concentration of H^+^ ions absorbed by liposomes made of phospholipids examined in this study and fraction 1 proteins is always higher than that of liposomes made of only phospholipids under the tested conditions.

## 3. Discussion

It has been proposed that the membranes of the myelin sheath and endoplasmic reticulum can absorb H^+^ ions on the membrane’s surface [40,42]. However, it has not been established which chemical species of a membrane, phospholipids or proteins, can absorb H^+^ ions. In this study, we isolated the five types of phospholipids commonly found in the membranes of the myelin sheath and endoplasmic reticulum, and we prepared unilamellar liposomes made of each type of phospholipid. We also isolated anionic proteins from the bee venom, and we prepared the five types of unilamellar liposomes made of each type of phospholipids, which were enriched with anionic proteins. The aim of this study was to confirm (1) that model membranes made of only phospholipids in the absence of proteins are capable of absorbing H^+^ ions, and (2) whether anionic proteins from bee venom, which supposedly evolved from common cellular membrane-active proteins [9,49,50], can increase the ability of membranes made of phospholipids commonly found in the myelin sheath and endoplasmic reticulum to absorb H^+^ ions from the bulk water. It should be emphasized that extrapolation from simplified phospholipid models to native myelin or ER membranes must be made cautiously. However, this is a study that provides a proof of concept that anionic phospholipids can increase the absorption of protons as observed in model phospholipid membranes; hence, future studies that examine the electrophysiological implications (increased electric capacitance and electrical conductivity) in model membranes and ex vivo, such as isolated ER, mitochondria, and other organelles, are warranted.

The molar concentration of H^+^ absorbed by a membrane was quantified from the difference in pH values of dd-H_2_O and sample of liposomes made of only phospholipids and liposomes made of phospholipids and anionic bee venom proteins. Notably, the standard deviation values derived from the triplicate pH measurements of samples prepared in triplicate for each data point were well below the total uncertainty induced by all pieces of equipment used in this study, which excludes the random errors during the experiments and allows us to conclude that the experimental data in this study are reliable. The very low ANOVA and *t*-test *p*-values confirm that the differences between the pH values of experimental data points derived from the different types of samples are statistically significant.

The results of this study show that pH in samples of all types of liposomes made of only phospholipids is higher than that in dd-H_2_O. The pH values in the liposomes made of both phospholipids and anionic proteins were higher than those in liposomes made of only phospholipids. This finding proves that anionic proteins increase the ability of membranes made of only phospholipids to absorb H^+^ ions. The amount of H^+^ ions absorbed by liposomes made of both phospholipids and proteins did differ for different types of phospholipids, which suggests that ability of membrane to absorb H^+^ ions depends on the types of intermolecular forces of attraction and repulsion between the polar moieties of phospholipid’s heads and amino acid residues of anionic proteins, which form the landscape of various functional groups on the membrane’s surface capable of absorbing H^+^ ions.

It should be noted that, surprisingly, in liposomes composed solely of phospholipids, the highest amount of H^+^ ions was absorbed by those made of neutral phosphatidylethanolamine (PE) rather than acidic phosphatidylserine or phosphatidylinositol. This phenomenon could be explained by the transformation of PE—a phospholipid with a high non-bilayer propensity—into inverted micelles upon interaction with H^+^ ions. This structural change leads to the engulfment of H^+^ ions within the inner hydrophilic volume of the non-bilayer inverted micelles, as described in previous studies [9,46,48,51,52].

Notably, in liposomes containing both phospholipids and proteins, the highest proton absorption was observed in those made of sphingomyelin (SM). This effect may be attributed to the presence of polar amino and alcohol groups in the sphingosine moiety of SM, which are notably absent in other phospholipids. It appears that these distinctive polar groups of SM’s sphingosine may offer greater opportunities for intermolecular bonding with the acidic proteins of bee venom when compared to the interactions possible with the glycerol backbone of other phospholipids. These conclusions collectively suggest an appealing duality in the behavior of SM’s polar groups. In pure phospholipid liposomes, the polar groups of SM’s sphingosine appear to be positioned deeper within the membrane architecture compared to the more superficially located glycerol backbones of other phospholipids. Conversely, in liposomes composed of both phospholipids and acidic proteins, the sphingosine polar groups of SMs demonstrate greater exposure toward the bulk water phase relative to the glycerol backbones of other phospholipids. This differential exposure ultimately provides SMs with enhanced capacity for proton absorption under these conditions. This observation supports the notion that anionic proteins interact in various fashions with phospholipids of different polar heads to generate different physico-chemical environments on membrane surfaces capable of absorbing different amount of H^+^ ions.

This study did not include non-anionic protein controls, which would be valuable in distinguishing the contributions of protein charge versus structural features. In addition, our reliance on bulk pH measurements represents an indirect proxy for membrane-bound proton accumulation. To this end, future studies should employ direct methods, such as pH-sensitive fluorescent dyes, infrared spectroscopy, or electrochemical probes, to more accurately quantify surface proton concentrations.

Overall, the findings of this study demonstrate that anionic proteins enhance the capacity of membranes composed of phospholipids found in myelin sheath and endoplasmic reticulum to absorb H^+^ ions. These results warrant further investigation to elucidate the amino acid sequences of these anionic proteins, which would enable in silico studies utilizing molecular dynamics simulations and AutoDock 4.2.6 version modeling. Such computational approaches would reveal conformational changes in the three-dimensional structure of anionic proteins as they transition from an aqueous environment to the membrane’s surface. Additionally, they would identify specific amino acid residues that engage in intermolecular interactions—including ionic interactions and ion-polar and hydrogen bonds—with the polar head groups of diverse phospholipids. These interactions would allow us to predict the collectively optimized membrane surface topography that would maximize the efficiency of proton absorption.

This line of research holds significant pharmacological relevance, as it provides novel insights into the bioenergetic mechanisms governing cellular energy accumulation and storage. A deeper understanding of these processes could facilitate the development of targeted pharmaceutical interventions to counteract the decline in cellular energy associated with pathological conditions and aging-related degeneration.

## 4. Materials and Methods

### 4.1. Materials

The following materials and chemicals were used in this study: Phospholipids—phosphatidylserine, sphingomyelin, phosphatidylinositol, phosphatidylcholine, and phosphatidylethanolamine—were purified from the rat liver (see the Preparations section below), Dichloro-diphenyl-trichloroethane (DDT), Sephadex G-25 and CM Sephadex C-50 (Nanjing Duly Biotech Co., Ltd., Nanjing, China), Tris(hydroxymethyl)aminomethane (Tris) 10 M pH 8.5 buffer, 1.0 M Tris-HCl pH 6.8 with 0.4% SDS buffer, Bromo-phenol Blue (Thomas Scientific, Swedesboro, NJ, USA); Mini-PROTEAN TGX precast gels (8% density), Isoelectric Focusing Gel Sample Buffer (IEF Gel), Ready Gel Precast Gels with ampholytes making pH gradient 3–10.5, 10× IEF Anode Buffer, 10× IEF Cathode Buffer (Bio-Rad Laboratories Co., Ltd., Shanghai, China), IEF p*I* 4.65–10.6 range protein markers for IEF (Shanghai Yeyuan Biotechnology Co., Ltd., Shanghai, China), Sodium Dodecyl Sulfate (SDS), 50× TAE (Tris-acetate-EDTA, pH 8.3) buffer, Coomassie Brilliant Blue-R-250, low-molecular-weight markers for SDS-PAGE (Thermo Fisher Scientific Inc., Shanghai, China), lyophilized bee venom (Sigma Aldrich, Saint Louis, MO, USA), 3.5 kDa cutoff dialysis tubing (Sigma Aldrich, Saint Louis, MO, USA), research grade Glycine (Asiamerica Group, Inc., Westwood, NJ, USA); Deionized-Distilled water (dd-H_2_O) (XiZhiMeng Co., Ltd., Shanghai, China). The rest of the chemicals, including methanol, glacial acetic acid, glycerol, KCl, and other common chemicals, were of reagent grade.

The following list of equipment was used in this study: Borosilicate 1.5 × 35 cm chromatography column (Cole-Parmer Instrument Co., LLC., Shanghai, China); Pierce™ Traut’s Reagent 2-iminothiolane Kit and *N*-succinimidyl 3-(2-pyridyldithio) propionate Kit (Thermo Fisher Scientific Inc., Carlsbad, CA, USA), UV-Visible range spectrophotometer JingHua 752 (Shanghai Jinghua Instruments Co., Ltd., Shanghai, China); Electronic high-precision analytical balance FA2004 (Shanghai Maiyi Ltd. Co., Shanghai, China); Vacuum freeze-drying lyophilizing machine YTLG-10 (Shanghai Yetuo Co., Ltd., Shanghai, China); Mini-PROTEAN Tetra Vertical Electrophoresis Cell for 2 Mini Precast Gels apparatus (Bio-Rad Laboratories Co., Ltd., Shanghai, China); Isoelectric Focusing (IEF) apparatus DYCP-37B (Beijing Liuyi Biotechnology Co., Ltd., Beijing, China); 125 V Power Generator MP3002D (Maisheng, Shanghai, China); TGL-16B microcentrifuge (Shanghai 3 Anting Co., Shanghai, China), Vacuum Pump TW-1M (Tingwei Co. Ltd., Wenling, China), Ultrasonic Dispenser Yt-JY96-IIN (Shanghai Yetuo technology Co., Ltd., Shanghai, China); pH Meter PHS-3C (Shanghai INESA Scientific Instruments Co., Ltd., Shanghai, China), Magnetic Stirrer JB-3 (Shanghai INESA Scientific Instruments Co. Ltd., China); Hamilton 100 microliter (µL) syringe (Hamilton Co., Boston, MA, USA).

### 4.2. Preparations

Column chromatography gel was prepared by placing 4 g of CM Sephadex C-50 powder into the 100 cm^3^ measuring cylinder and pouring into the same cylinder 10 mmol/L Tris pH 8.5 buffer to bring the volume to 60 mL. The gel slurry was incubated for 5 h at 25 °C and then placed gently into the 1.5 × 35 cm borosilicate chromatography column. The starting buffer was 10 M Tris pH 8.5.

The sample used for column chromatography was 1.0 g of bee venom dissolved in 1.5 mL starting buffer.

The gradient buffer used for column chromatography was 0.5 KCl in 10 M Tris pH 8.5 (starting buffer). It was prepared by placing 2.725 g KCl in 100 mL volumetric flask and pouring the starting buffer to 100 mL.

The SDS-PAGE sample buffer was prepared by placing 0.2 g SDS, 10 mg bromophenol blue, 0.25 mL 1.0 mol/dm^3^ DTT, 1.0 mL glycerol, and 0.5 mL 1.0 M Tris-HCl pH 6.8 with 0.4% SDS buffer into 10 mL volumetric flask and bringing volume to 10 mL with dd-H_2_O.

Protein samples for SDS-PAGE were prepared by placing, using a Hamilton syringe, the 25 µmL sample buffer into eight 1.5 mL Eppendorf plastic vials, and then placing 25 µmL of low-molecular-mass markers and 25 µmL of each of the seven bee venom fractions (Figure 1) at a protein concentration of 2 µg/µL into separate 1.5 cm^3^ Eppendorf plastic vials, which were then boiled for 5 min and centrifuged at 12,000× *g* for 2 min. In total, 20 µL supernatants from Eppendorf plastic vials were loaded using a Hamilton syringe into the precast gel wells.

The SDS-PAGE running buffer was prepared by placing 10 mL 50× TAE buffer, 7.2 g glycine, and 0.2 g SDS into a 500 mL volumetric flask and pouring dd-H_2_O into the flask to make a 500 mL volume.

Protein samples for IEF were prepared by placing 5 mg of lyophilized protein into a 5 mL volumetric flask and bringing the volume to 5 mL with dd-H_2_O to make 1.0 mg/mL of protein solution.

Anode IEF buffer was prepared by mixing 2 mL 10× IEF Anode Buffer with 18 mL dd-H_2_O. 

Cathode IEF buffer was prepared by mixing 2 cm^3^ 10× IEF Cathode Buffer with 18 cm^3^ dd-H_2_O.

The staining solution for SDS-PAGE and IEF was prepared by dissolving 0.5 g Coomassie brilliant blue-R-250 in 75 mL methanol and 25 mL glacial acetic acid, and then adding 150 mL dd-H_2_O.

The distaining solution for SDS-PAGE and IEF was prepared by 100 mL mixing methanol, 25 mL glacial acetic acid, and 125 mL distilled water.

Animal housing and husbandry: Male Wistar rats were housed in groups of 5–10 in solid-bottom polycarbonate cages with stainless-steel wire-bar lids for food and water provision. The cages contained absorbent bedding (corn cob or paper-based chips) for environmental enrichment and waste absorption. Adult male rats exceeding 400 g were allocated a minimum of 450 square inches of floor space.

Euthanasia protocol: Euthanasia was induced by placing the animal in a dedicated chamber and introducing CO_2_ at a flow rate of 40% of the chamber volume per minute. After loss of consciousness was confirmed, death was ensured via decapitation by highly trained personnel.

Phospholipids were isolated from the rat liver in the Institute of Cell Biophysics of the Russian Academy of Sciences, Pushchino, Russia. Livers were excised from three euthanized, 6-month-old male Wistar rats (average body weight 450 g), yielding a combined mass of 45 g. The tissue was immediately placed in a Petri dish containing ice-cold 0.88% NaCl solution to facilitate the dissection and removal of loose connective tissue and fat. The liver mass was homogenized in a Waring Blender 7011G (Conair Corporation, McMinnville, TN, USA) for 3 min in the mixture of 300 mL chloroform and 200 mL methanol. Then blending continued for another minute in 100 mL of methanol and 200 mL dd-H_2_O. The homogenate was centrifuged on Sorvall Bios A centrifuge for 10 min at 200× *g*. The aqueous layer of methanol and dd-H_2_O was discarded. The chloroform phase containing lipids was washed with 400 mL 0.88% NaCl solution by centrifuging the mixture for 10 min at 200× *g*. The 0.88% NaCl aqueous layer was discarded. The washing procedure with 400 mL dd-H_2_O was repeated two more times. The chloroform phase with lipids was filtrated through Whatman No. 1 filter paper on a funnel. Chloroform was removed from the filtrate by drying with the LAV-3 2 Stage Laboratory High Vacuum Pump for 60 min. The total lipid extract was dissolved in 30 mL methanol and 20 mL dd-H_2_O and then purified on silica column. The individual phospholipids—phosphatidylserine, sphingomyelin, phosphatidylinositol, phosphatidylcholine, and phosphatidyl-ethanolamine—were isolated by the immunoaffinity columns containing the resin Sephadex G-25 covalently linked to the antibodies specific to the polar heads of either phosphatidylserine, sphingomyelin, phosphatidylinositol, phosphatidylcholine, or phosphatidylethanolamine. Covalent linking was completed by using Pierce™ Traut’s Reagent 2-iminothiolane Kit and *N*-succinimidyl 3-(2-pyridyldithio) propionate Kit (Thermo Fisher Scientific Inc., Carlsbad, CA, USA). Antibodies were developed by injecting the synthetic polar heads of the above phospholipids into the adjuvant-primed adult male Wistar rats. Ten days after injection, ascites fluid was collected from the peritoneal cavity of rats, and the antibodies from ascites fluid were purified on a Protein-A-Sepharose column. The fractions of individual phospholipids eluted from the immunoaffinity columns were collected in the centrifuge tubes with 20 mL chloroform. The fractions in chloroform were centrifuged for 10 min at 200× *g*, after which the aqueous layer was discarded. Chloroform in the centrifuge tubes was removed by drying for 45 min with the LAV-3 2 Stage Laboratory High-Vacuum Pump. The mass of phospholipid in the centrifuge tube was determined as the difference in the mass of the centrifuge tube with phospholipid and the empty centrifuge tube. The number of moles of phospholipids in the centrifuge tube was assessed using the following formula: mass of phospholipid in the centrifuge tube ÷ molar mass of phospholipid. A phospholipid sample in each tube was dissolved in 20 mL chloroform, which was then transferred to 50 mL measuring cylinder, and an additional volume of chloroform was added to adjust the concentration of phospholipid in the chloroform to 0.1 M. The phospholipid samples in the chloroform were then sealed in 50 mL containers and were used as the stock phospholipid samples.

The phospholipid solution in chloroform for making lipid–protein liposomes was prepared by placing 100 µL of 0.1 M phospholipid solution in chloroform using a Hamilton syringe to the 10 mL volumetric flask and adding chloroform to the same flask to make a total volume of 10 mL.

The aqueous solution derived from of fraction 1 proteins employed for preparing lipid-protein liposomes was prepared by placing 0.125 mg of protein to the 10 mL volumetric flask and adding dd-H_2_O to the same flask to make a total volume of 10 mL. The average molecular mass of protein from fraction 1 was estimated using SDS-PAGE to be 12,500 g/mol (Figure 2).

### 4.3. Methods

The bee venom (1 g) dissolved in the starting buffer was applied onto the chromatography column. The eluant was collected in the 3 mL portions in different tubes, and gradient buffer was applied at tube number 23 (Figure 1). The presence of protein in eluant was assessed spectrophotometrically by reading optical density at 280 nm. The eluant was pooled into the seven fractions, as shown in the chromatogram in Figure 1. All seven fractions were dialyzed at room temperature overnight against 2 L dd-H_2_O using the 3.5 kDa cutoff dialysis bags. The dd-H_2_O was changed four times, and the bulk water was continuously stirred with the magnetic stirrer. Dialyzed protein fractions were kept at −40 °C for five hours and then lyophilized at 2 Pa vacuum and −70 °C for 10 h by using the vacuum freeze-drying lyophilizing machine YTLG-10.

The protein from seven fractions and low-molecular-mass markers were prepared the SDS-PAGE runs, as described above. The Mini-PROTEAN Tetra Vertical Electrophoresis Cell was filled with the 500 mL SDS-PAGE running buffer, and the Mini-PROTEAN TGX precast gel (8% density) was mounted in one chamber of the Cell and the blank plastic dam was mounted in another chamber of the Cell. In total, 20 µL of low-molecular-mass markers and 20 µL of each protein fraction were loaded into the wells of precast gel. The Electrophoresis Cell was connected to the 125 V power generator, and the electrophoretic run proceeded at 100 V for about 50 min. The progress of the proteins traveling in the gel was monitored by observing the movement of the bromophenol blue frontline. After the completion of electrophoretic run, the precast gel was rinsed with dd-H_2_O, and the gel was gently separated from the plastic cast. The gel was first incubated in staining solution for 30 min and then incubated in distaining solution for about 60 min until protein bands in the gel become clearly visible.

The horizontal plate of the Isoelectric Focusing apparatus DYCP-37B was cleaned with dd-H_2_O and dried with filter paper, and the Ready Gel Precast Gel with pH range 3–10.5 was mounted on the clean horizontal plate. The electrodes for IEF run were prepared by soaking the two filter paper strips, one in the Anode IEF buffer and another in the Cathode IEF buffer, and placing the filter strips on the opposite sides of the Precast Gel and mounting on the filter strips the electrode wires of the Isoelectric Focusing apparatus DYCP-37B, which was then connected to the 125 V power generator at 75 V for 30 min. The 4 µL of the IEF p*I* 4.65–10.6 range protein markers and 5 µL of each of the seven protein samples for IEF were applied on the gel’s surface in the middle of the Precast Gel. The power generator was turned on at 90 V for about 120 min, and the traveling of venom proteins in the gel was monitored by observing the traveling of the pre-stained IEF protein markers. After the completion of the IEF run, the Precast Gel was placed to the staining solution for 30 min. The IEF gel was detached from the plastic support cast and the gel was incubated for another 30 min. The gel was then incubated in the distaining solution for about 90 min until the protein bands became clearly visible.

Sonicated unilamellar liposomes made of phospholipids and venom fraction 1 were made by placing 100 µL of phospholipid solution in chloroform for making lipid–protein liposomes in the 5 mL glass tube. Chloroform from the phospholipid solution was removed by 30 min vacuum treatment using Vacuum Pump TW-1M to form a lipid film. The aqueous solution of fraction 1 proteins for making lipid–protein liposomes (100 µL) was diluted in 5 mL dd-H_2_O and placed onto the lipid film, which was then sonicated with the Ultrasonic Dispenser Yt-JY96-IIN at 22 kHz for 15 min to form unilamellar liposomes. The liposomes were then transferred to the 10 mL volumetric flask, where the total volume was brought to 10 mL by adding dd-H_2_O. The concentration of phospholipids and protein in liposomes was 10^–5^ and 10^–8^ mol/L, respectively. The liposomes without fraction 1 proteins were made as described above, without the addition of fraction 1 proteins into the 5 mL glass tube. The pH measurements in dd-H_2_O and in phospholipid liposome suspensions with and without fraction 1 proteins were made using the pH Meter PHS-3C. Prior to pH measurements, liposome samples with and without fraction 1 proteins were incubated for 1 h at 37 °C to allow for protons in dd-H_2_O and those bound to the liposome surface to reach equilibrium.

### 4.4. Uncertainties

Electronic analytical balance ± 0.0001 g: (0.0001 g/7.2 g) × 100% = 0.0014%;

250 cm^3^ volumetric flask ± 0.1000 cm^3^: (0.1000 cm^3^/150 cm^3^) × 100 = 0.03667%;

100 cm^3^ volumetric flask ± 0.1000 cm^3^: (0.1000 cm^3^/60 cm^3^) × 100% = 0.1667%;

50 cm^3^ volumetric flask ± 0.0500 cm^3^: (0.0500 cm^3^/25 cm^3^) × 100% = 0.2000%;

10 cm^3^ volumetric flask ± 0.0100 cm^3^: (0.0100 cm^3^/10 cm^3^) × 100% = 0.1000%;

100 µdm^3^ Hamilton syringe: (0.1000 µdm^3^/25 µdm^3^) × 100% = 0.4000%;

PH meter ± 0.0100 pH units: (0.0100/8.23) × 100% = 0.1215%;

Digital thermometer (±0.0500 °C): (0.0500 °C/25.5 °C) × 100% = 0.1960%;

Total uncertainty: 1.2223%.

### 4.5. Statistical Analysis

For all pH readings, the samples of liposomes were prepared in triplicate for each data point. The standard deviation between the three readings was never higher 0.3% of the means. We used an ANOVA test to compare the means among six groups of pH readings, including dd-H_2_O and liposomes made of five different phospholipids. The *t*-test was used to compare the means of pH readings between liposomes made of phospholipids and liposomes made of phospholipids with proteins. A statistically significant difference was considered when *p* value was less than 0.05 (*p* < 0.05).

### 4.6. Safety Measures

DDT and chloroform are toxic if swallowed, inhaled, or in contact with skin and eyes, and are carcinogenic when ingested [53,54]. Methanol is flammable and causes skin burns and eye damage, and is toxic when swallowed or inhaled, and may cause death when ingested [55]. As a safety measure, all experiments with DDT, chloroform, and methanol were conducted under a fume hood. The open flame was kept away from methanol. While working in the lab, investigators wore a laboratory coat, safety goggles, and latex gloves.

### 4.7. Ethical Issues

The adult male Wistar rats used in this study are not an endangered or protected species. Animal handling and experimental procedures were conducted according to the Instruction NO. 57 of 30 December 2011 for the use of laboratory animals of the Biological Safety and Ethics Committee of the Institute of Cell Biophysics of the Russian Academy of Sciences, which fully agrees with the Directive 2010/63/EU of the European Parliament.

## 5. Conclusions

The findings of this study demonstrate that anionic proteins significantly enhance the capacity of phospholipid membranes, examined here under the tested conditions, to absorb and retain protons—a property critical for membrane-associated energy storage. This mechanistic insight supports the emerging paradigm that extramitochondrial membranes, such as those of the myelin sheath and endoplasmic reticulum, can function as proton capacitors, independently of classical chemiosmotic gradients. The observed synergy between anionic proteins and phospholipids—particularly sphingomyelin—suggests that tailored lipid–protein complexes could optimize proton absorption efficiency.

These results hold substantial pharmacological promise for addressing diseases linked to cellular energy deficits, such as neurodegenerative disorders (e.g., Alzheimer’s and Parkinson’s diseases), mitochondrial myopathies, and age-related metabolic decline. By elucidating the molecular determinants of proton absorption, this work provides a foundation for designing small-molecule mimetics or peptide-based therapeutics that replicate or augment the proton-capturing function of anionic membrane proteins. For instance, synthetic analogs of bee venom proteins could be engineered to stabilize proton gradients in compromised cellular membranes, thereby restoring bioenergetic efficiency. Alternatively, modulating the lipid composition of cellular membranes—via dietary supplements or lipid nanoparticles—might enhance endogenous proton storage capacity in tissues reliant on extramitochondrial energy production, such as neurons and myelin.

Future research should prioritize the structural characterization of the protein–phospholipid interfaces responsible for proton absorption, enabling rational drug design. In vivo studies are also needed to validate whether enhancing membrane proton storage can ameliorate pathological energy deficits. Collectively, this work opens a new avenue for developing therapies that target membrane-based bioenergetic mechanisms, offering potential alternatives to conventional mitochondrial-centric approaches.

## Figures and Tables

**Figure 1 pharmaceuticals-18-01334-f001:**
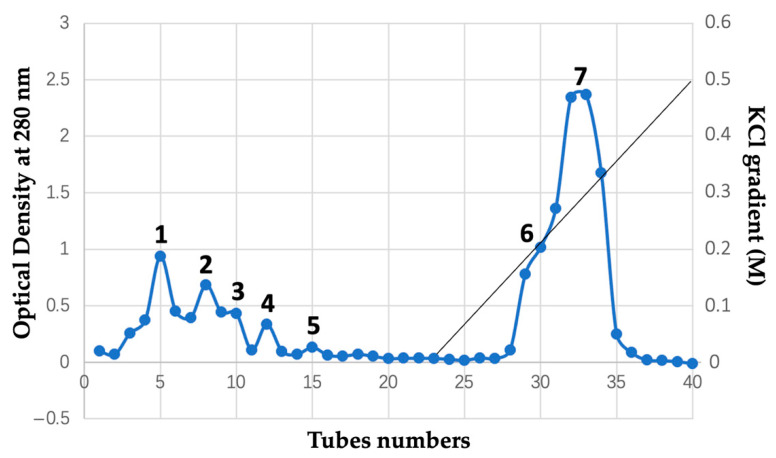
CM Sephadex C-50 cation exchange column chromatography of crude bee venom. Chromatogram is a function of optical density of eluant taken at 280 nm over the tube numbers where eluant was collected in volumes of 3 mL per tube. The tubes were pooled in seven fractions, as shown. Starting buffer is 10 mM Tris-buffer, pH 8.5. Gradient buffer is a starting buffer with 0.5 M KCl. Application of gradient buffer started from tube 23, as indicated by the straight line. Column 1.5 × 35 cm. Sample: 1.0 g bee venom.

**Figure 2 pharmaceuticals-18-01334-f002:**
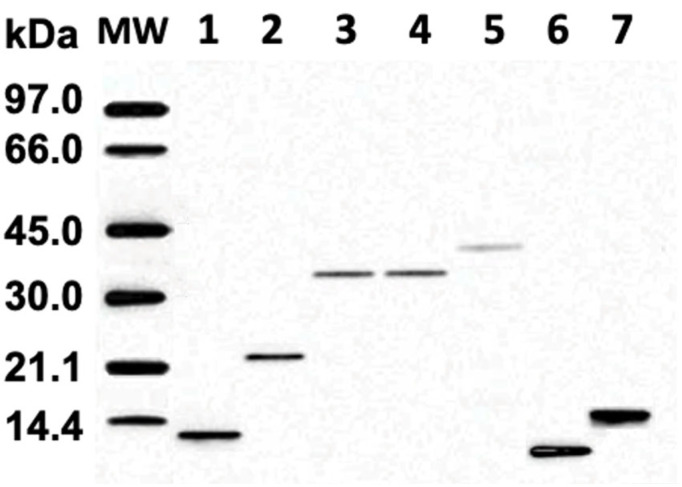
SDS-polyacrylamide 8% gel electrophoresis of bee venom fractions 1–7. Fraction 1 includes tube 3–6, fraction 2 includes tubes 7–9, fraction 3 includes tubes 10–11, fraction 4 includes tubes 12–13, fraction 5 includes tubes 14–16, fraction 6 includes tubes 27–30, and fraction 7 includes tubes 31–36, as shown in Figure 1. Lane MW carries the standard low-molecular-weight markers.

**Figure 3 pharmaceuticals-18-01334-f003:**
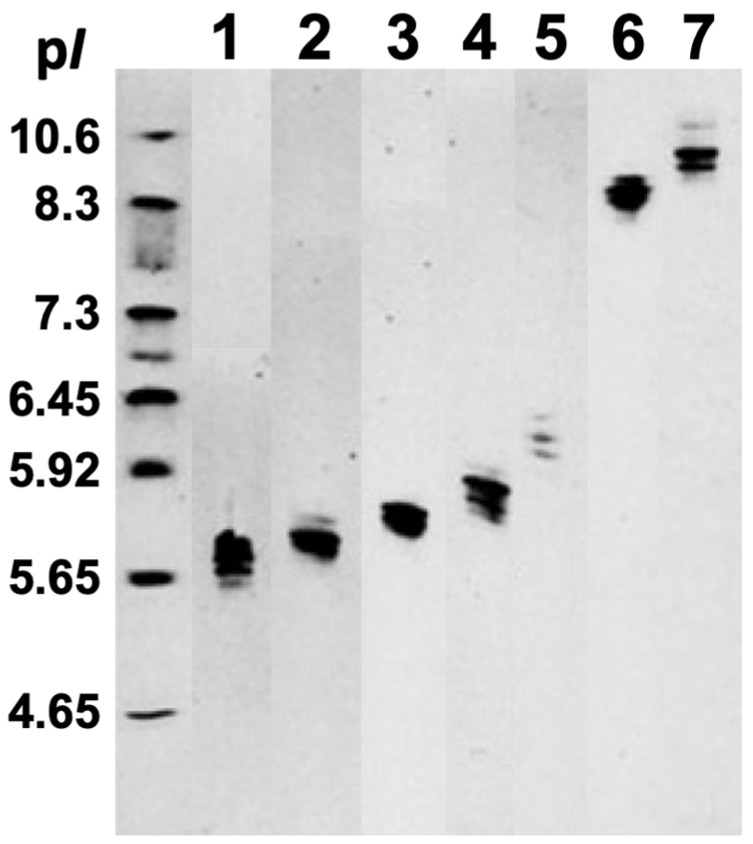
Isoelectic focusing (IEF) of the bee venom fractions 1–7 in Ready Gel Precast Gels with ampholytes of pH gradient from 3.0 to 10.5. A multi-sample horizontal IEF gel was employed with 11 lanes containing the seven bee venom fractions and four unrelated samples. For presentation in Figure 3, the lanes of the seven bee venom fractions were excised from the original IEF gel photograph (Appendix A) and are arranged adjacently to facilitate comparative analysis. Lane 1 includes tube 3–6, lane 2 includes tubes 7–9, lane 3 includes tubes 10–11, lane 4 includes tubes 12–13, lane 5 includes tubes 14–16, lane 6 includes tubes 27–30, and lane 7 includes tubes 31–36, as shown in Figure 1. The standard p*I* markers (FMC Corporation, Rockland, ME, USA) sample of a p*I* range from 4.65 to 10.6 is applied on the unmarked lane on the far left.

**Figure 4 pharmaceuticals-18-01334-f004:**
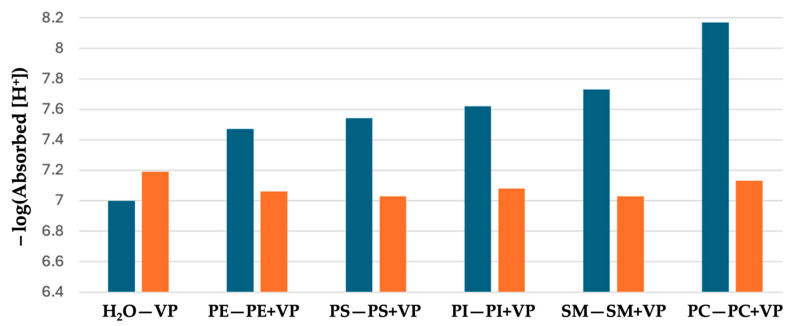
Negative logarithm of [H^+^] absorbed by fraction 1 proteins (VP) in dd-H_2_O and by membrane of liposomes made of either phospholipids only or phospholipids with VP. Emerald bars: H_2_O—dd-H_2_O; PE—phosphatidylethanolamine; PS—phosphatidylserine; PI—phosphatidylinositol; SM—sphingomyelin; PC—phosphatidylcholine. Orange bars: VP—fraction 1 proteins in dd-H_2_O; the remaining bars represent membranes of liposomes made of phospholipids with VP.

**Table 1 pharmaceuticals-18-01334-t001:** Optical density (OD) values of eluant from the CM Sephadex C-50 cation exchange column chromatography of crude bee venom taken at 280 nm in volumes of 3 mL per tube. Eluting buffer 10 mM Tris-buffer, pH 8.5. Gradient buffer is eluant buffer with 0.5 M KCl. Application of gradient buffer started from tube 23.

Tube Number	OD 280 nm	Tube Number	OD 280 nm	Tube Number	OD 280 nm	Tube Number	OD 280 nm
1	0.097	11	0.108	21	0.035	31	1.360
2	0.069	12	0.335	22	0.035	32	2.343
3	0.258	13	0.096	23	0.033	33	2.366
4	0.371	14	0.071	24	0.026	34	1.674
5	0.937	15	0.131	25	0.016	35	0.250
6	0.453	16	0.063	26	0.038	36	0.085
7	0.395	17	0.053	27	0.034	37	0.019
8	0.683	18	0.070	28	0.108	38	0.014
9	0.444	19	0.052	29	0.778	39	0.005
10	0.429	20	0.032	30	1.016	40	0.000

**Table 2 pharmaceuticals-18-01334-t002:** The pH readings of the samples prepared in triplicate for each data point taken at 25 °C in dd-H_2_O and solutions of liposomes made of either phosphatidylethanolamine, phosphatidylserine, phosphatidyl-inositol, sphingomyelin, or phosphatidylcholine. Also given in Table 2 are the mean values of the pH readings, standard deviations (SDs), and the ANOVA *p* value. The concentration of phospholipids in all samples is 10^–5^ M.

	The pH Values	Means	SD	ANOVA *p* Value
dd-H_2_O	7.01 7.00 6.99	7.00	0.0082	3.71 × 10^−8^
Phosphatidylethanolamine	7.16 7.19 7.19	7.18	0.0141	
Phosphatidylserine	7.17 7.14 7.14	7.15	0.0141	
Phosphatidylinositol	7.11 7.11 7.14	7.12	0.0141	
Sphingomyelin	7.09 7.08 7.10	7.09	0.0082	
Phosphatidylcholine	7.02 7.05 7.02	7.03	0.0141	

**Table 3 pharmaceuticals-18-01334-t003:** Comparison of statistical difference in the pH values between samples of liposomes made of only phospholipids and liposomes made of phospholipids and fraction 1 proteins. The concentrations of phospholipids and fraction 1 proteins are 10^–5^ M and 10^–8^ M, respectively.

**Phosphatidylethanolamine Liposomes (PE) Against PE + Fraction 1 Proteins**				
	**The pH Values**	**Means**	**SD**	***t*-Test *p* Value**
PE	7.16 7.19 7.19	7.18	0.0141	9.42 × 10^–7^
PE + fraction 1 proteins	7.87 7.90 7.90	7.89	0.0141	
**Phosphatidylserine Liposomes (PS) Against PS + Fraction 1 Proteins**				
	**The pH Values**	**Means**	**SD**	***t*-Test *p* Value**
PS	7.17 7.14 7.14	7.15	0.0141	2.40 × 10^–7^
PS + fraction 1 proteins	8.17 8.14 8.14	8.15	0.0141	
**Phosphatidylinositol Liposomes (PI) Against PI + Fraction 1 Proteins**				
	**The pH Values**	**Means**	**SD**	***t*-Test *p* Value**
PI	7.11 7.11 7.14	7.12	0.0141	2.40 × 10^–6^
PI + fraction 1 proteins	7.76 7.78 7.74	7.76	0.0163	
**Sphingomyelin liposomes (SM) Against SM + Fraction 1 Proteins**				
	**The pH Values**	**Means**	**SD**	***t*-Test *p* Value**
SM	7.09 7.08 7.10	7.09	0.0082	1.58 × 10^–8^
SM + fraction 1 proteins	8.23 8.22 8.24	8.23	0.0082	
**Phosphatidylcholine Liposomes (PC) Against PC + Fraction 1 Proteins**				
	**The pH Values**	**Means**	**SD**	***t*-Test *p* Value**
PC	7.02 7.05 7.02	7.03	0.0141	2.43 × 10^–6^
PC + fraction 1 proteins	7.58 7.61 7.58	7.59	0.0141	

**Table 4 pharmaceuticals-18-01334-t004:** The initial concentrations of H^+^ ions before adding liposomes and/or fraction 1 proteins, final concentrations of H^+^ ions after adding liposomes and/or fraction 1 proteins, and concentration of H^+^ ions absorbed by either fraction 1 proteins, liposomes, or liposomes with fraction 1 proteins. Concentrations of phospholipids and fraction 1 proteins are 10^–5^ M and 10^–8^ M, respectively.

	Initial [H^+^] M	Final [H^+^] M	Absorbed [H^+^] M
Fraction 1 proteins in dd-H_2_O	1 × 10^−7^	3.55 × 10^−8^	6.45 × 10^−8^
Phosphatidylethanolamine (PE)	1 × 10^−7^	6.61 × 10^−8^	3.39 × 10^−8^
PE + fraction 1 proteins	1 × 10^−7^	1.29 × 10^−8^	8.71 × 10^−8^
Phosphatidylserine (PS)	1 × 10^−7^	7.08 × 10^−8^	2.92 × 10^−8^
PS + fraction 1 proteins	1 × 10^−7^	7.08 × 10^−9^	9.29 × 10^−8^
Phosphatidylinositol (PI)	1 × 10^−7^	7.59 × 10^−8^	2.41 × 10^−8^
PI + fraction 1 proteins	1 × 10^−7^	1.74 × 10^−8^	8.26 × 10^−8^
Sphingomyelin (SM)	1 × 10^−7^	8.13 × 10^−8^	1.87 × 10^−8^
SM + fraction 1 proteins	1 × 10^−7^	5.89 × 10^−9^	9.41 × 10^−8^
Phosphatidylcholine (PC)	1 × 10^−7^	9.33 × 10^−8^	6.70 × 10^−9^
PC + fraction 1 proteins	1 × 10^−7^	2.57 × 10^−8^	7.43 × 10^−8^

## Data Availability

The data presented in this study are available in this article.

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
