# Peer review of "Bee Venom Proteins Enhance Proton Absorption by Membranes Composed of Phospholipids of the Myelin Sheath and Endoplasmic Reticulum: Pharmacological Relevance"

_pharmaceuticals, 2025, doi:10.3390/ph18091334_

Round 1
Reviewer 1 Report
Comments and Suggestions for Authors
Revision of article 3825278
The article "Bee Venom Proteins Enhance Proton Absorption by Membranes Composed of Phospholipids of Myelin Sheath and Endoplasmic Reticulum: Pharmacological Relevance" measures the degree of proton leakage from liposomes with a phospholipid composition similar to that of myelin by the addition of a protein extracted from bee venom. This supports both the hypothesis of myelin's role as a proton capacitor and the toxicity of the bee venom protein.
The experimental measurements are well conducted, and both the introduction and conclusions are well-developed.
Suggeste Minor Revision
In the Discussion-Conclusion, one could mention the role played by cationic proteins such as Myelin Basic Protein in strengthening the role of proton capacitor played by myelin, while the protein isolated from bee venom is, on the contrary, anionic and therefore would disturb the function of the proton capacitor. The topic of the role of the proton capacitor played by myelin in sleep is developed by the article:
Myelin: A possible proton capacitor for energy storage during sleep and energy supply during wakefulness.
Prog Biophys Mol Biol. 2025 Mar 28;196:91-101. www.doi.org/10.1016/j.pbiomolbio.2025.03.001. PMID: 40157615.
which could be cited in this context.
Minor observation: Line 56-57. The phrase “driving the rotation of its catalytic site and enabling ATP synthesis” should be changed to “driving the rotation of Fo subunit resulting in ATP synthesis by the F1 subunit”
Author Response
Reviewer 1 comments authors answers
The article "Bee Venom Proteins Enhance Proton Absorption by Membranes Composed of Phospholipids of Myelin Sheath and Endoplasmic Reticulum: Pharmacological Relevance" measures the degree of proton leakage from liposomes with a phospholipid composition similar to that of myelin by the addition of a protein extracted from bee venom. This supports both the hypothesis of myelin's role as a proton capacitor and the toxicity of the bee venom protein.
The experimental measurements are well conducted, and both the introduction and conclusions are well-developed.
Suggeste Minor Revision
In the Discussion-Conclusion, one could mention the role played by cationic proteins such as Myelin Basic Protein in strengthening the role of proton capacitor played by myelin, while the protein isolated from bee venom is, on the contrary, anionic and therefore would disturb the function of the proton capacitor. The topic of the role of the proton capacitor played by myelin in sleep is developed by the article:
Myelin: A possible proton capacitor for energy storage during sleep and energy supply during wakefulness.
Prog Biophys Mol Biol. 2025 Mar 28;196:91-101. www.doi.org/10.1016/j.pbiomolbio.2025.03.001. PMID: 40157615.
which could be cited in this context.
Authors answer: We have included the above reference to describe the term proton capacitor as a form of energy storage to support various activities during wakefulness for example.
Minor observation: Line 56-57. The phrase “driving the rotation of its catalytic site and enabling ATP synthesis” should be changed to “driving the rotation of Fo subunit resulting in ATP synthesis by the F1 subunit”
Authors answer: We have changed the phrase “driving the rotation of its catalytic site and enabling ATP synthesis” to “driving the rotation of Fo subunit resulting in ATP synthesis by the F1 subunit” – see lines 58-59.

Reviewer 2 Report
Comments and Suggestions for Authors
Comments regarding the manuscript
Bee Venom Proteins Enhance Proton Absorption by Membranes Composed of Phospholipids of Myelin Sheath and Endoplasmic Reticulum: Pharmacological Relevance.
Zhuoyan Zeng, Mingsi Wei, Shuhao Zhang, Hanchen Cui, Ruben K. Dagda, Edward S. Gasanoff
Snake venom, despite its toxicity, is a valuable source for developing various medicines, including those for cardiovascular conditions and cancer. Similarly, bee venom has a long history of use in traditional medicine, including anti-inflammatory and pain-relieving properties, it has demonstrated potential in neurodegenerative diseases, including Parkinson's disease. Bee venom is studied for its potential anticancer effects. It may inhibit cancer growth, trigger apoptosis and even enhance the effects of chemotherapy in certain cancers. Melittin is a peptide found in honeybee venom, known for its potent membrane-disrupting and anti-inflammatory properties. It's a major component of bee venom, representing approximately 50% of its dry mass. So, investigation of bee venom proteins certainly is interesting for Pharmaceuticals.
This paper demonstrates that liposomes from different lipids absorb H+ ions from purified water, and this effect is increased in the presence of bee venom proteins with an isoelectric point near 5.7. As a result, for example, in the presence of phosphatidyl serin liposomes with proteins pH of the aqueous solution increased to 8.15. Similar effects may be important in membrane-based bioenergetics processes.
My questions/suggestions to the authors:
- Even purified water still has some buffer capacity, for example, due to dissolved from air CO2. It would make sense to measure it, and to use method of standard editions of an acid to see exactly how many protons were absorbed.
- Eluting buffer 10 mM Tris-buffer, pH 8.5 was used in Sepharose columns. What would be the result if it were, for example, 5.5 when the proteins are not charged? How does conversion to K+ form and then lyophilization influence the final state of the proteins and the results?
- You did experiments with liposomes in water, proteins in water, sonicated liposomes with protein. What if you do it with sonicated liposomes first, and then add the protein? How fast will the system reach the equilibrium pH?
- Would it make sense to take absorbed H+ concentration, subtract what is absorbed by pure liposomes, and then divide by protein concentration, 10-8 . You will get nearly 9 in all cases.
- electric potential localized to the membrane interface- line 65.
Conclusion: I would recommend that the paper be published after the authors address these comments and questions.
Author Response
Reviewer 2 comments & authors answers
Bee Venom Proteins Enhance Proton Absorption by Membranes Composed of Phospholipids of Myelin Sheath and Endoplasmic Reticulum: Pharmacological Relevance.
Zhuoyan Zeng, Mingsi Wei, Shuhao Zhang, Hanchen Cui, Ruben K. Dagda, Edward S. Gasanoff
Snake venom, despite its toxicity, is a valuable source for developing various medicines, including those for cardiovascular conditions and cancer. Similarly, bee venom has a long history of use in traditional medicine, including anti-inflammatory and pain-relieving properties, it has demonstrated potential in neurodegenerative diseases, including Parkinson's disease. Bee venom is studied for its potential anticancer effects. It may inhibit cancer growth, trigger apoptosis and even enhance the effects of chemotherapy in certain cancers. Melittin is a peptide found in honeybee venom, known for its potent membrane-disrupting and anti-inflammatory properties. It's a major component of bee venom, representing approximately 50% of its dry mass. So, investigation of bee venom proteins certainly is interesting for Pharmaceuticals.
This paper demonstrates that liposomes from different lipids absorb H+ ions from purified water, and this effect is increased in the presence of bee venom proteins with an isoelectric point near 5.7. As a result, for example, in the presence of phosphatidyl serin liposomes with proteins pH of the aqueous solution increased to 8.15. Similar effects may be important in membrane-based bioenergetics processes.
My questions/suggestions to the authors:
Even purified water still has some buffer capacity, for example, due to dissolved from air CO2. It would make sense to measure it, and to use method of standard editions of an acid to see exactly how many protons were absorbed.
- Eluting buffer 10 mM Tris-buffer, pH 8.5 was used in Sepharose columns. What would be the result if it were, for example, 5.5 when the proteins are not charged? How does conversion to K+ form and then lyophilization influence the final state of the proteins and the results?
Authors’ response: We thank the reviewer for this thoughtful question regarding the use of CM Sephadex C-50 in cation exchange chromatography. The purpose of this technique in our study was to purify positively charged proteins (pI > 8.5) by exploiting their ability to bind to the negatively charged resin. As the K⁺ ion concentration increases during the gradient, these cationic proteins are displaced from the resin and elute in later fractions (e.g., tubes 28–29, as shown in Figure 1).Proteins that are uncharged at the buffer pH (pI ≈ 7.0) would not bind to the column under these conditions and thus co-elute with anionic proteins in the early fractions. In our experiments, fractions 1–5 represent anionic proteins, whereas fractions 6–7 reflect cationic proteins that were displaced during the K⁺ gradient. Importantly, the conversion of the column to the K⁺ form and subsequent lyophilization of protein fractions are not expected to alter the intrinsic charge state of the proteins. Upon resolubilization at pH 7.0, the proteins reassume their native charge characteristics, as confirmed by our isoelectric focusing experiments (5 mg lyophilized protein in 5 mL water, pH 7.0), which corroborated that cationic proteins retained pI values >7.0 in agreement with the chromatography results.
1. You did experiments with liposomes in water, proteins in water, sonicated liposomes with protein. What if you do it with sonicated liposomes first, and then add the protein? How fast will the system reach the equilibrium pH?
Authors’ response: We appreciate the suggestion to sonicate liposomes first and then add protein. In our protocol, we form a lipid film, hydrate it in the presence of protein, and then sonicate the protein–lipid mixture to generate proteoliposomes (Methods, lines 454–460). Co-sonication (or co-reconstitution) is the standard approach because protein insertion into pre-formed, sonicated liposomes is inefficient and typically yields surface adsorption rather than bona fide membrane incorporation. Allowing the bilayer to self-assemble around the protein during sonication supports correct partitioning and stable proteoliposome formation.
Regarding equilibration kinetics: bulk pH reflects free protons after rapid mixing and partitioning between the aqueous phase and membrane/protein binding sites. In practice, bulk pH equilibrates quickly (mixing-limited), and in our measurements we allowed a defined equilibration period prior to recording and observed no further drift within the instrument’s resolution over the subsequent monitoring window. We will clarify this in the Methods by stating the equilibration interval and stability criterion used. Taken together, co-sonication ensures protein incorporation, and the observed stable pH after equilibration indicates that the reported values represent equilibrium conditions for the tested compositions.
2. Would it make sense to take absorbed H+ concentration, subtract what is absorbed by pure liposomes, and then divide by protein concentration, 10-8 . You will get nearly 9 in all cases.
Authors’ response: We thank the reviewer for this thoughtful suggestion. In our experiments, however, the triplicate pH recordings already reflect the concentration of free protons remaining in bulk solution across liposomes, proteoliposomes, and aqueous controls. Because these measurements capture the equilibrium between protons bound by both phospholipids and proteins, dividing the observed pH difference by protein concentration would not accurately isolate the contribution of proteins to proton absorption. Instead, the most direct and appropriate comparison is the difference in bulk pH (and thus proton concentration) across the experimental conditions, as presented in our study. We agree that complementary approaches such as electrophysiological recordings could further dissect protein-specific contributions to membrane charge and potential, and we plan to pursue these in future studies.
3. Electric potential localized to the membrane interface- line 65.
Authors’ response: Yes, this phrase is correctly written as noted in line 65. It reflects the experimentally validated concept, demonstrated through electrophysiological recordings of isolated proteoliposomes, that anionic phospholipids within the lipid bilayer can act as proton carriers and thereby confer an overall positive electric potential to the membrane.
Conclusion: I would recommend that the paper be published after the authors address these comments and questions.

Reviewer 3 Report
Comments and Suggestions for Authors
In this work, Zeng et al. consider the effect of Bee Venom Proteins on the absorption of protons in proxies for myelin sheath and the ER. The manuscript makes some large claims that are largely unsubstantiated. If the authors heavily redact the manuscript to remove all unjustified claims (below) then it could proceed to be published:
Comments:
- The central hypothesis—that bee venom proteins enhance proton absorption in a way that is pharmacologically relevant— needs to be justified further. The authors extrapolate from in vitro liposome models to broad claims about energy storage in myelin and ER membranes without sufficient mechanistic or physiological evidence. In particular, the authors state that “Five phospholipids (phosphatidylethanolamine, phosphatidylserine, phosphatidylinositol, sphingomyelin, phosphatidylcholine) from rat liver were isolated to model myelin/ER membranes”. How valid is this approximation? This needs to be justified significantly.
- The term “proton capacitor” is used repeatedly with references [41-44] mostly by Morelli. Can you please supply some papers that are not by Morelli to substantiate this terminology in the community? Otherwise it remains a metaphor rather than a quantifiable biophysical concept.
- Methodological Weaknesses: I have two points here:
- Your experimental design lacks critical controls. For example, there is no comparison with non-anionic proteins, nor any attempt to isolate the effect of protein charge versus structure.
- The use of pH changes in bulk water as a proxy for proton absorption is oversimplified and potentially misleading. No direct measurement of membrane-bound proton concentration is provided.
- The authors state: “Overall, our results demonstrated that all liposomal membranes absorb protons from bulk water, and that liposomes enriched with anionic proteins exhibit greater proton absorption than their protein-free counterparts. These findings underscore the capacity of lipid-protein membranes to accumulate protons at their surface, supporting a revised view of membrane-associated energy storage.”
Either revise this statement to be “that the liposomal membranes considered in this study did X” or provide proofs at all liposomal membranes do this.
There are many grammatical omissions that need to be addressed:
- Page 2 line 71: “Groundbreaking work of Morelli group provided evidence of extramitochondrial 71 (in absence of mitochondria)” should be “Groundbreaking work of the Morelli group provided evidence of extramitochondrial 71 (in absence of mitochondria)”. Please refrain from terms like groundbreaking, unless a Nobel prize or the like was awarded.
- Page 2 line 72: et al should be et al.
- “chemi-osmotic” → should be “chemiosmotic”
- “concertation” → should be “concentration”
- “logariphm” → should be “logarithm”
- “memebrane” (in abbreviations) → should be “membrane”
- “Petry dish” → should be “Petri dish”
- “Warning Blender” → likely meant “Waring Blender”
- “polled into the seven fractions” → should be “pooled into seven fractions”
- “overage molecular mass” → should be “average molecular mass”
- “Te IEF gel” → should be “The IEF gel”
- “Although, the binding…” → remove the comma after “Although”
- “This could be explained by the insertion…” → consider rephrasing for clarity: “This may be due to the insertion…”
- “leading to the reduction in concentration of absorbed H⁺ ions” → better phrased as “resulting in reduced proton absorption”
- “the rest of bars represent membrane…” → should be “the remaining bars represent membranes…”
- Use consistent formatting for p-values (e.g., p < 0.001 vs. p-value = 2.43 × 10⁻⁶).
- Ensure consistent use of abbreviations (e.g., dd-H₂O vs. ddH₂O).
- Clarify whether “liposomes” refer to unilamellar or multilamellar vesicles throughout.
Author Response
Reviewer 3 comments & authors answers
In this work, Zeng et al. consider the effect of Bee Venom Proteins on the absorption of protons in proxies for myelin sheath and the ER. The manuscript makes some large claims that are largely unsubstantiated. If the authors heavily redact the manuscript to remove all unjustified claims (below) then it could proceed to be published:
Comments:
1. The central hypothesis—that bee venom proteins enhance proton absorption in a way that is pharmacologically relevant— needs to be justified further. The authors extrapolate from in vitro liposome models to broad claims about energy storage in myelin and ER membranes without sufficient mechanistic or physiological evidence. In particular, the authors state that “Five phospholipids (phosphatidylethanolamine, phosphatidylserine, phosphatidylinositol, sphingomyelin, phosphatidylcholine) from rat liver were isolated to model myelin/ER membranes”. How valid is this approximation? This needs to be justified significantly.
Authors answer: We thank the reviewer for pointing out the need to justify the central hypothesis and the use of isolated phospholipids to model myelin and ER membranes. Our study is exploratory and uses well-established biophysical models (liposomes) to probe fundamental properties of phospholipids and their interactions with proteins. While isolated phospholipids cannot fully replicate the complexity of native myelin or ER membranes, they are commonly used to study lipid–protein interactions in a controlled manner. We have revised the Introduction and Discussion to clarify the limitations of our model and to frame our findings as preliminary insights that may inspire further physiological studies rather than direct extrapolations to in vivo conditions.
2. The term “proton capacitor” is used repeatedly with references [41-44] mostly by Morelli. Can you please supply some papers that are not by Morelli to substantiate this terminology in the community? Otherwise it remains a metaphor rather than a quantifiable biophysical concept.
Authors answer: We agree that relying exclusively on the work of Morelli et al. may give the impression that the 'proton capacitor' concept is not widely recognized. To address this, we have added additional references that discuss surface proton binding, lateral proton conduction, and the role of membranes as proton reservoirs (e.g., Haines & Dencher, 2002; Heberle, 2000; Toczyłowska-Mamińska, 2017; Kocherginsky, 1979, 2009). We now present 'proton capacitor' explicitly as a metaphorical framework used by Morelli and colleagues, while acknowledging that the underlying phenomenon—surface proton accumulation—is supported by a broader literature.
3. Methodological Weaknesses: I have two points here:
-
- Your experimental design lacks critical controls. For example, there is no comparison with non-anionic proteins, nor any attempt to isolate the effect of protein charge versus structure.
- The use of pH changes in bulk water as a proxy for proton absorption is oversimplified and potentially misleading. No direct measurement of membrane-bound proton concentration is provided.
Authors answer: We appreciate the reviewer’s concerns regarding experimental design and measurement limitations. (1) Regarding controls: We did not include non-anionic proteins in this preliminary study, focusing instead on whether anionic proteins could enhance proton absorption. We acknowledge that future work should include non-anionic and structurally unrelated proteins to distinguish charge-specific from structural effects. (2) Regarding the use of bulk pH changes: We recognize that pH changes in bulk water represent an indirect measure of proton absorption. While this method has been used in prior liposome studies, it does not provide direct quantification of membrane-bound protons. We have revised the Discussion to state this limitation and to propose future use of direct techniques such as pH-sensitive fluorescent dyes, IR spectroscopy, or electrochemical probes to more precisely measure surface-bound proton concentrations.
4. The authors state: “Overall, our results demonstrated that all liposomal membranes absorb protons from bulk water, and that liposomes enriched with anionic proteins exhibit greater proton absorption than their protein-free counterparts. These findings underscore the capacity of lipid-protein membranes to accumulate protons at their surface, supporting a revised view of membrane-associated energy storage.”
Either revise this statement to be “that the liposomal membranes considered in this study did X” or provide proofs at all liposomal membranes do this.
Authors answer: We thank the reviewer for pointing out this overgeneralization. We have revised the manuscript to clarify that our conclusions apply specifically to the liposomal membranes examined in this study, and we have avoided extrapolating to 'all liposomal membranes' without qualification.
Comments on the Quality of English Language
There are many grammatical omissions that need to be addressed:
- Page 2 line 71: “Groundbreaking work of Morelli group provided evidence of extramitochondrial 71 (in absence of mitochondria)” should be “Groundbreaking work of the Morelli group provided evidence of extramitochondrial 71 (in absence of mitochondria)”. Please refrain from terms like groundbreaking, unless a Nobel prize or the like was awarded.
Authours answer: We have changed term Groundbreaking to Innovative.
- Page 2 line 72: et al should be et al.
- “chemi-osmotic” → should be “chemiosmotic”
- “concertation” → should be “concentration”
- “logariphm” → should be “logarithm”
- “memebrane” (in abbreviations) → should be “membrane”
- “Petry dish” → should be “Petri dish”
- “Warning Blender” → likely meant “Waring Blender”
- “polled into the seven fractions” → should be “pooled into seven fractions”
- “overage molecular mass” → should be “average molecular mass”
- “Te IEF gel” → should be “The IEF gel”
- “Although, the binding…” → remove the comma after “Although”
- “This could be explained by the insertion…” → consider rephrasing for clarity: “This may be due to the insertion…”
- “leading to the reduction in concentration of absorbed H⁺ ions” → better phrased as “resulting in reduced proton absorption”
- “the rest of bars represent membrane…” → should be “the remaining bars represent membranes…”
- Use consistent formatting for p-values (e.g., p < 0.001 vs. p-value = 2.43 × 10⁻⁶).
- Ensure consistent use of abbreviations (e.g., dd-H₂O vs. ddH₂O).
Authors answer: We have corrected all grammatical omissions noted by the reviewer above.
- Clarify whether “liposomes” refer to unilamellar or multilamellar vesicles throughout.
Authors answer: In this study we have used unilamellar sonicated liposomes as indicated in the Materials and Methods.

Round 2
Reviewer 3 Report
Comments and Suggestions for Authors
The authors have satisfied my concerns.